



# Effects of moisture absorption on damage progression and strength of unidirectional and cross-ply fiberglass-epoxy composites

Jake D. Nunemaker, Michael M. Voth, David A. Miller[1], Daniel D. Samborsky, Paul Murdy, Douglas S. Cairns,

[1]Mechanical and Industrial Engineering Department, Montana State University, Bozeman, MT, 59717, USA

*Correspondence to*: David A. Miller (davidmiller@montana.edu)

**Abstract.** Fiber-Reinforced-Polymer composites (FRP's) possess superior mechanical properties and formability, making them a desirable material for construction of large optimized mechanical structures, such as aircraft, wind turbines, and marine hydro kinetic (MHK) devices. However, exposure to harsh marine environments can result in moisture absorption into the microstructure of the FRP's comprising these structures and often degrading mechanical properties. Specifically, laminate static and fatigue strengths are often significantly reduced, which must be considered in design of FRP structures in marine environments. A study of fiber-glass epoxy unidirectional and cross-ply laminates was conducted to investigate hygrothermal effects on the mechanical behavior of a common material system used in wind applications. Several laminates were aged in 50°C distilled water until maximum saturation was reached. Unconditioned control and the saturated samples were tested in quasi-static tension with the accompaniment of Acoustic Emission (AE) monitoring. Cross-ply laminates experienced 54% reduction in strengths from due to moisture absorption, while unidirectional laminates strengths were reduced by 40%. Stress-strain curves and AE activity of the samples were analyzed to identify changes in damage progression due to aging.

## 1 Introduction

### 1.1 Composites and renewable energy technologies

FRP's offer desirable properties for development of large mechanical structures, such as wind turbines and more recently, Marine-Hydro-Kinetic (MHK) devices. High specific strength and stiffness, the ability to tailor the anisotropic properties, and low costs make FRP's the primary choice for optimizing the design of energy harvesting devices (Samborsky et al., 2012). Additionally, the formability of FRP's enables the manufacture of complex geometry and structures necessary in the Wind and MHK industry. As wind energy becomes a more dominant energy source, and as MHK technology progresses, it is paramount to understand how FRP's perform throughout a device's designed lifetime. Inherently, this means characterizing material degradation from environmental exposure. This research explores how exposure to a wet environment affects the mechanical response of Fiber-Glass Epoxy laminates. Effort is made to characterize the effect of moisture on composites on a coupon level, which can provide insight to subsequent component and structure design.

### 1.2 Hygrothermal aging of Fiber Reinforced Plastics

FRP's typically exhibit Fickian diffusion, primarily through the matrix material as fiberglass typically absorbs a negligible amount of moisture. Other modes are possible, but defect driven; poorly bonded fiber-matrix interfaces and porosity provide



these secondary diffusion pathways (Sun et al., 2011;Tsenoglou et al., 2006). At ambient design temperatures moisture diffusion occurs over time periods impractical for study in the laboratory setting. The rate-temperature relationship of the Arrhenius law applies for moisture diffusion in FRP's (Tsai et al., 2009), and allows test specimens to be aged at elevated temperatures to increase diffusion rates. Elevated aging temperatures are chosen to increase diffusion rates without

degrading the temperature-sensitive matrix material. Exceedingly high temperatures can alter the material physically or chemically as evidenced by non-Fickian diffusion and/or changes in glass transition temperature (Tsai et al., 2009;Grammatikos et al., 2016).

Past research has shown significant reductions in both static and fatigue performance due to moisture absorption(Mourad et al., 2010;Miller et al., 2012;Liao et al., 1999;Komai et al., 1991;Siriruk and Penumadu, 2014;Nunemaker, 2017, 2016).

These changes are consequence of changes in the mechanical behavior of the matrix and quality of the fiber/matrix interface. Water is a known plasticizer in polymers; water molecules disrupt interactions between polymer chains resulting in increased chain mobility and ultimately leading to degraded mechanical properties. Moisture induced plasticization has been observed experimentally through changes in viscoelastic properties and glass-transition temperature of neat resin (Zhou and Lucas, 1999a, b;Nogueira et al., 2001). Interface integrity is more challenging to determine experimentally, but scanning electron

microscope (SEM) images of fracture surfaces of aged composites have provided evidence of a weakened interface bond (Mourad et al., 2010;Liao et al., 1999).

Understanding matrix and interface response to hygrothermal aging is essential to better understanding of how moisture absorption affects FRP's at a micromechanical level. However, the changes in matrix and the interface properties can be difficult to translate to the mechanical response of the laminate and do not fully describe the macroscopic damage

progression of the material. Thus, laminate scale testing is still necessary to characterize mechanical behavior of these materials in hygrothermal environments. Unidirectional tension and transverse tension are commonly used to characterize the change in mechanical response due to hygrothermal aging (Miller et al., 2012). These tests can be applied to predict behavior of more complex laminates; however, this approach does not incorporate hygrothermal effects on ply interactions. This research expands this idea to experimentally examine cross-ply laminates and unidirectional laminas in both

unconditioned and fully conditioned states, to investigate the effects of hygrothermal aging on the mechanical response and damage progression of multi-angle laminates.

## 1.3 Composites and Acoustic Emission

Damage progression at the micromechanical level is difficult to experimentally observe. *In situ* Acoustic Emission monitoring (AE) is a technique that enables real-time observation of damage events using piezo-electric sensors mounted on

the surface of the test specimen to record damage initiated elastic waves propagating through the material. These waveforms can provide useful information for both structural health monitoring (SHM) and material characterization purposes. For this



research, material characterization is the focus, with the intent of using acoustic events to help characterize changes in damage progression due to environmental aging.

AE is an indirect method of measurement and therefore requires thorough analysis to correlate AE activity to micromechanical damage within the material. Many analysis techniques exist including single parameter analysis (Bourchak et al., 2007;Suzuki et al., 1987;deGroot et al., 1995;Ramirez-Jimenez et al., 2004), clustering and artificial neural networks (Gutkin et al., 2011;Suresh Kumar et al., 2017;Pashmforoush et al., 2012) , as well as detailed waveform analysis (Ni and Iwamoto, 2002;Surgeon and Wevers, 1999;Voth, 2017). In this study, single parameter methods of Fast-Fourier-Transform (FFT) peak frequency and event energy are used to characterize the damage progression of the material. Past research has attempted to correlate FFT-peak frequencies in AE waveforms to specific damage mechanisms (deGroot et al., 1995;Ramirez-Jimenez et al., 2004;Suzuki et al., 1987); while, energy has been shown to correlate with damage accumulation in the material (Bourchak et al., 2007;Kumar et al., 2017). When used as parameters for ANN, energy and other parameters indicative of damage event magnitude, such as counts and number of hits, have successfully been used to predict residual strength (Suresh Kumar et al., 2017). Although multi-variable techniques permit relationships to be made between many AE parameters in the analysis, differentiating between damage mechanisms remains a challenge. Clustering studies have shown that frequency content of AE waveforms persists as a dominant parameter for differentiating types damage events (Gutkin et al., 2011;Pashmforoush et al., 2012).

Previous research incorporating acoustic emission monitoring in the evaluation of hygrothermally-aged composites has revealed that moisture uptake frequently causes a decrease in AE response in terms of energy, amplitude, number of events, and counts (Assarar et al., 2011;Czigány et al., 1995;Garg and Ishai, 1985a;Garg and Ishai, 1985b;Hamstad, 1983;Komai et al., 1991). However, this may not always be the case: a recent delamination study showed an increase in magnitude of AE response in terms of energy and amplitude for conditioned delamination samples (Liu et al., 2017). Changes in the magnitude of the AE response due to moisture have not been thoroughly explored or understood, but have been theorized to be an effect of matrix plasticization and its effect on signal attenuation and damage source behavior. Consideration of this change in AE response is important when analyzing AE results, and can help differentiate changes in AE response associated from damage behavior from effects of moisture on the AE propagation behavior.

## 2 Experimentation

### 2.1 Materials

Vectorply E-LT3800, a stitched E-glass fabric consisting of 1138 g/sq. m (91%) unidirectional tows and 114 g/sq. m (9%) transverse backing fibers, was utilized in this study. The architecture represents reinforcements commonly used in wind turbine blades. Although commonly treated as a unidirectional fabric, the 9% backing fibers can have notable effects on laminate behavior, as discussed later. Four laminates, $[0]_2$, $[90]_2$, $[0/90]_s$, and $[90/0]_s$, were tested in this study. The $[0]_2$ and $[90]_2$ laminates were chosen to be representative of individual 0° and 90° lamina. Both $[0/90]_s$ and $[90/0]_s$ were tested to



compare variation due to stacking sequence. Epikote RIMR 135 epoxy resin with Epicure RIMH1366 hardener, manufactured by Hexion was used as the resin system. Laminates were manufactured using vacuum assisted resin infusion (VARI), and cured at 22°C for 24 hours followed by a 12 hour post cure at 70°C as per the resin specifications. Resulting fiber volume fractions varied between 55% and 58%. Test specimens were cut to 30mm x 300mm with a diamond abrasion

saw. Tabs measuring 64mm x 30mm were adhered to all test specimens, as seen in Figure 2, to reduce effects of the gripping pressure throughout tensile testing.

**2.2 Hygrothermal Aging**

Samples were hygrothermally aged by immersion in distilled water maintained at 50°C. A scale with a 0.001 gram resolution was used to measure the initial mass of the samples and subsequent mass changes due to moisture uptake. The

bulk moisture uptake, *M(t),* is given by Equation 1, where $m_t$ is the mass of the sample during moisture sorption and $m_i$ is the mass prior to absorption.

$$M(t) = \frac{m_t - m_i}{m_i},$$
(1)

Test specimens reached saturation when the bulk moisture content reached equilibrium, seen by the plateau in the bulk diffusion curves shown in Figure 1. The saturated moisture content, $m_\infty$, ranged between 0.89% and 0.94%, and was

consistent between layups. Fully conditioned samples are referred to as "saturated" and control samples with no aging, are referred to as "dry".

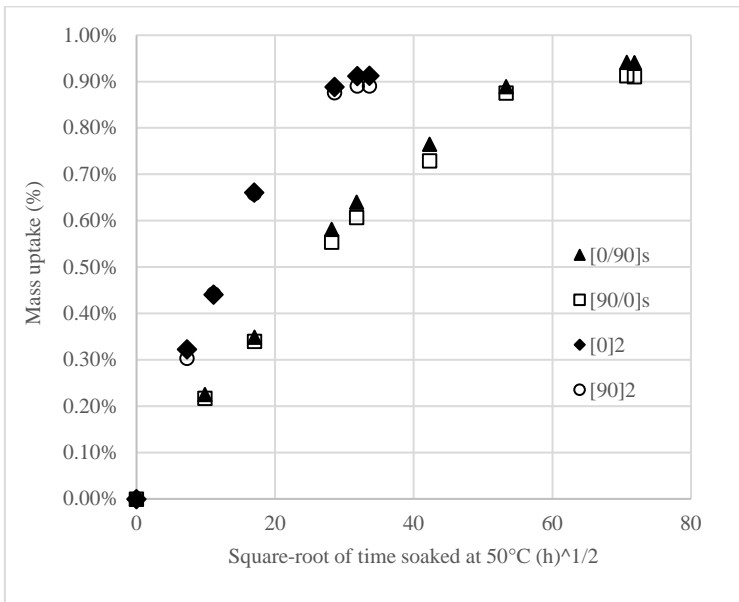

**Figure 1: Moisture Uptake Curves**



### 2.3 Testing

### 2.3.1 Tensile Testing

Five tensile tests were conducted for each laminate in both the dry and saturated conditions, in accordance with ASTM D3039 (2014).Testing took place using an Instron 8562 100kN servo-mechanical load frame with a crosshead speed of 0.06

inches/min. An Instron 2620-824 extensometer captured strain across a 0.5" gage section. Both dry and saturated samples were tested to failure in ambient conditions.

### 2.3.2 Acoustic Emission Setup and Analysis

All tests were conducted with in situ Acoustic Emission monitoring, using a Mistras PCI-8 Micro-II Samos system and two Physical Acoustic WD sensors. The sensors were connected in line with 40dB external preamplifiers and featured a 50-

1000kHz operating range. A linear array was used to record acoustic activity throughout the tensile tests. The sensors were spaced 128mm apart, each 64mm from the longitudinal center of the coupon as seen in Figure 2.  A thin layer of vacuum grease was applied to ensure proper acoustic contact between the sensor and the coupon.

The MISTRAS system operated at a sampling rate of 3 MHz. Waveforms were collected with 128k of pretrigger and a total

waveform length of 1024k. A band-pass filter of  50kHz to 400kHz to eliminate noise of undesired frequencies and ensure the collected data fell in the operating range of the sensors (2011). The timing parameters peak definition time (PDT), hit definition time (HDT), hit lockout time (HLT), and max duration were set to 50, 100, 300, and 99 microseconds, respectively. An AE system functionality check and a pencil lead break test were conducted prior to each test to ensure proper system set up and physical contact between the coupon and sensors.


During post processing, acoustic activity was truncated beyond the maximum load. This was for two reasons: damage after maximum load is not pertinent to damage progression in the material and extensive damage present in the sample significantly affects event propagation and attenuation. The latter may also affect AE events immediately preceding final failure, which is an important consideration when evaluating final damage progression.



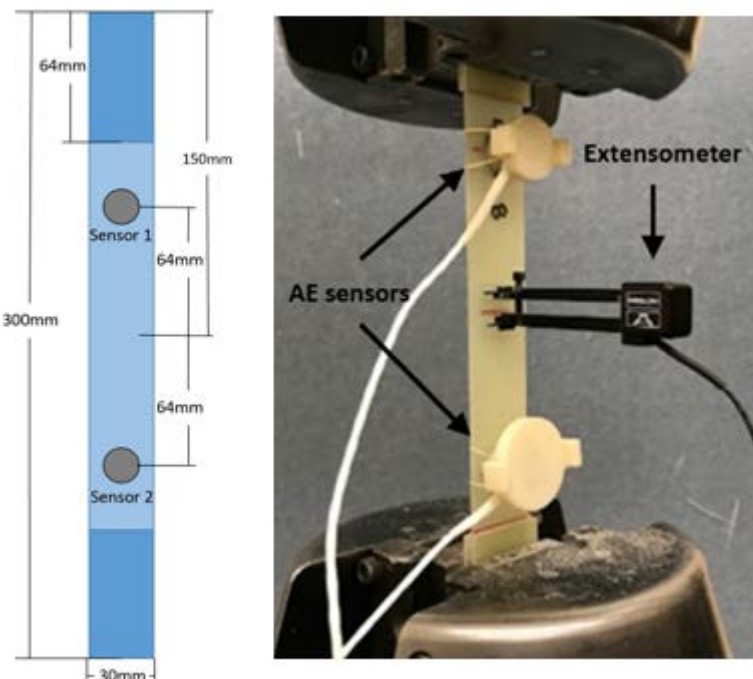

Figure 2: Coupon layout geometry and sensor layout (left). Experimental test setup (right).

## 3 Results

### 3.1 Mechanical Properties

5    Average ultimate strengths of the cross-ply laminates and unidirectional laminates in each environmental condition are displayed in Figure 3. Standard deviation error bars are added to each test group. All laminates experienced significant moisture induced strength reductions. Cross-ply laminate strengths were reduced by 54%, while the $[0]_2$ and $[90]_2$ laminates experienced 39% and 41% strength reductions respectively.




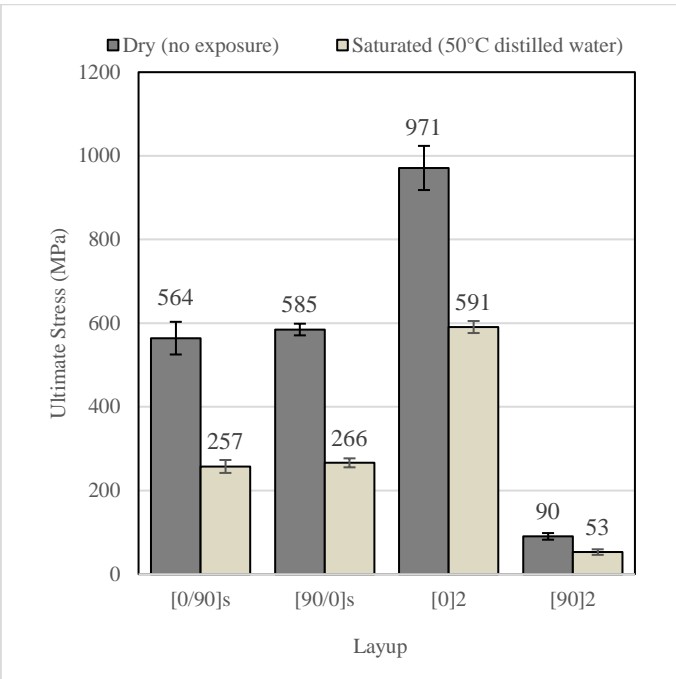

**Figure 3: Average ultimate stresses of dry and saturated samples**

However, it is important to note that comparing changes in strengths of unidirectional laminates to cross-ply laminates does not fully describe the lamina level changes. Thus, ultimate unit-loads were also used to compared laminates since the 90° lamina add little strength in laminates with 0° plies. This allows a more direct comparison of changes in load bearing 0° plies. These ultimate loads are shown in Figure 4. In the dry condition, the cross-ply laminates failure loads reflect the sum of the $[0]_2$ and $[90]_2$ loads. Conventionally, adding layers to a composite increases the load carrying capacity of the laminate regardless of orientation. However, in the saturated state, the converse was true; the cross-ply laminates attained considerably lower ultimate loads than the $[0]_2$ laminate alone. The cross-ply laminates exhibited lower ultimate loads after saturation than the $[0]_2$ laminates themselves.



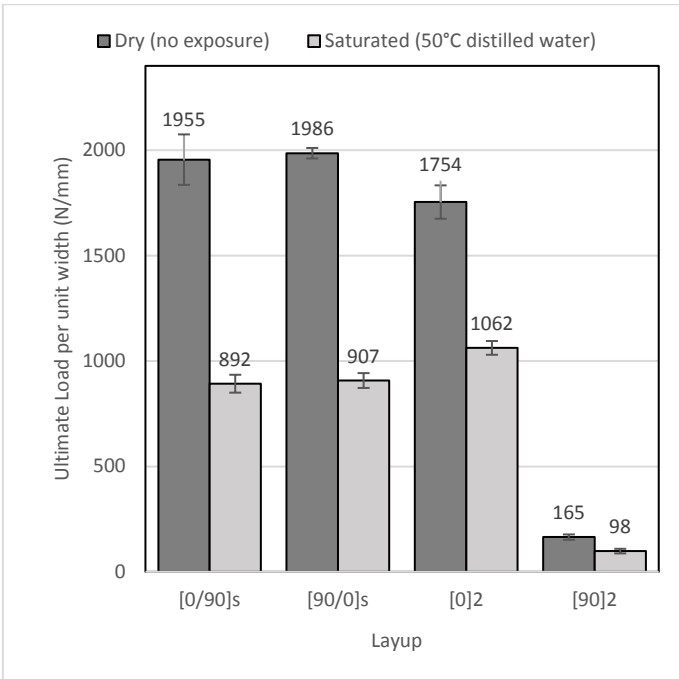

**Figure 4: Ultimate loads per unit width of dry and saturated samples**

The typical stress-strain response for dry and saturated laminates are displayed in Figure 5. The $[90]_2$ laminates show a clear change in damage progression: the knee in the $[90]_2$ stress-strain curve, causing a bilinear response, correlates to onset of damage, predominantly transverse cracking, occurring at 0.43% strain in dry specimens and 0.28% strain in saturated specimens. The dry $[90]_2$ also feature a final increase in stiffness, causing a trilinear response, which is a result of load transfer to the small amount longitudinally-oriented backing strands of the fabric; although backing strands are still present, this behavior does not occur in the saturated specimens. The stress-strain response of the saturated $[0]_2$ is nearly identical to that of the dry condition, however truncated at a lower ultimate stress and strain.





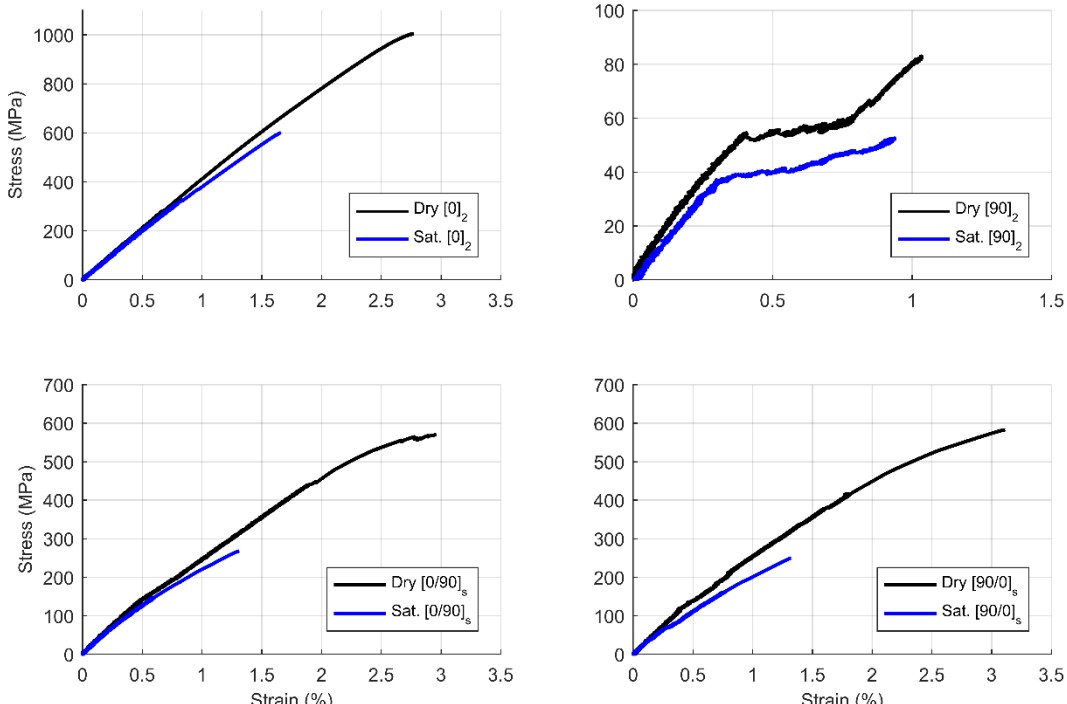

**Figure 5: Stress-Strain response of dry and saturated laminates**

The stress-strain behavior of the $[0]_2$ and $[90]_2$ translates to the cross-ply laminates. The cross-ply laminates follow a
response similar to the $[0]_2$. Both the dry and saturated samples initially follow similar stress-strain curves, but the saturated
samples fail at much lower strain levels. This similarity is to be expected as the $0^o$ lamina act as the primary load bearing
plies within the cross-ply laminates. Interestingly, there is slight deviation in the response between dry and saturated cross-
plies at about 0.3% strain, which correlates to the lower damage initiation strain in the saturated $[90]_2$. The similar response
of both cross-ply laminates suggests the behavior was independent of stacking sequence.

## 3.2 Coupon Failures

Failed coupons were inspected to provide insight into changes of damage progression with moisture absorption. Images were
captured with a high-resolution flat-bed scanner or DSLR camera. Inspection of failed $[90]_2$ specimens revealed a change in
crack density with aging. Figure 6 shows the crack spacing of a dry and saturated $[90]_2$ sample. Red ink was applied as a
penetrant to enhance the contrast of the cracks. The crack density in the dry sample (Fig 6 lower) is notably higher, with the
cracks being uniformly spaced in correlation with each transverse tow. On the contrary, the saturated $[90]_2$ (Fig. 6 upper)
failed at a lower crack density with a more inconsistent crack spacing.





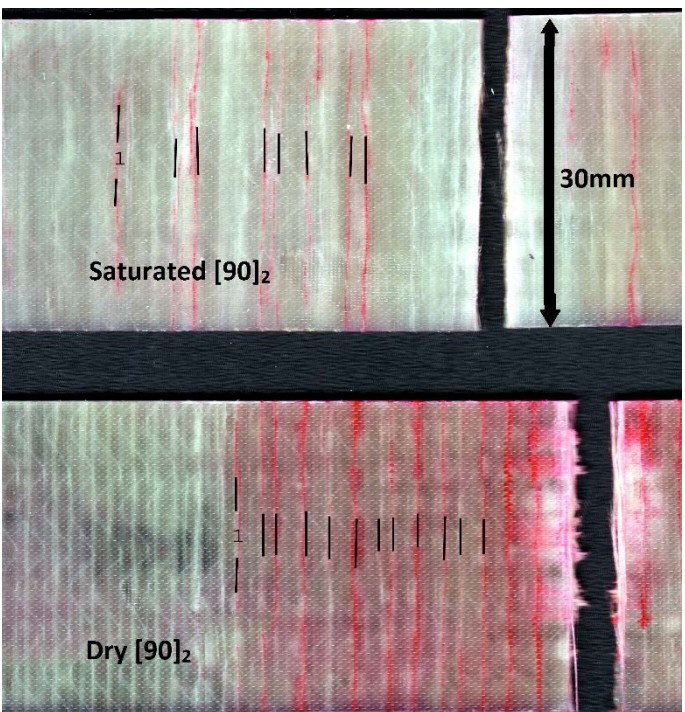

**Figure 6: Transverse cracking in failed [90]₂ samples (black lines mark transverse cracks)**

Failed $[0]_2$, $[0/90]_s$, and $[90/0]_s$ coupons are displayed in Figures 7, 8, and 9, respectively. Dry coupons containing 0° plies
failed at a higher rate of strain energy release than their saturated counterparts, resulting in explosive, wide spread damage.
This damage is illustrated by the complete tow separation in the $[0]_2$ (Fig.7 upper), the brooming of the 0° ply in the $[0/90]_s$
(Fig. 8 upper), and the large delamination of the 90° ply in the $[90/0]_s$ (Fig. 9 upper). In the saturated $[0]_2$ (Fig. 7 lower),
fiber tows failed in several locations resulting in damage dispersed throughout the coupon, but exhibited very little brooming
effect.

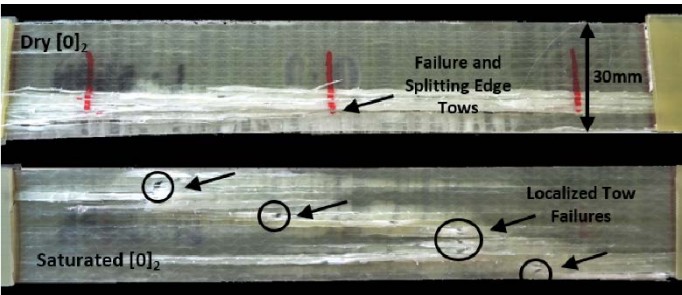

**Figure 7: Failed [0]₂ coupons dry (top) saturated (bottom)**





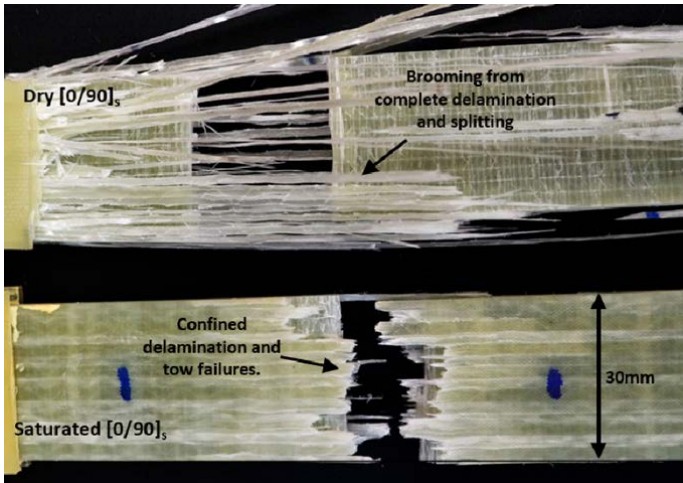

**Figure 8: Failed [0/90]ₛ coupons dry (top) saturated (bottom)**

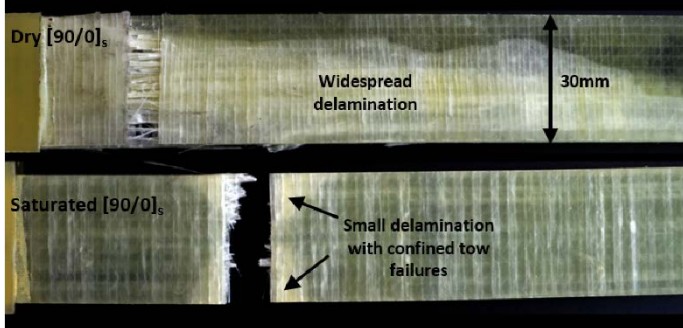

**Figure 9: Failed [90/0]ₛ coupons dry (top) saturated (bottom)**

In the saturated cross-ply coupons, damage was much more localized producing a neat transverse failure. This is shown in both the [0/90]$_s$ (Fig. 8 lower) and [90/0]$_s$ (Fig. 9 lower) where the 0° tow failures occur in close proximity to each other within the gage section. This phenomenon was consistent throughout the dataset.

**3.3 Acoustic Emission**

**3.3.1 FFT Peak Frequency Analysis**

The FFT peak frequency of each damage event was plotted against strain to illustrate the damage progression throughout loading. FFT peak frequency results for unidirectional and cross-ply laminates are shown in Figures 10 and 11. Saturated samples accumulate significantly fewer events than the respective dry samples. Despite fewer events, dry and saturated

samples of the same layups exhibit similar prevalent frequencies throughout the progression of damage.



However, a notable difference between damage onset as evident in the AE response. The acoustic events begin to occur at lower strain levels in the saturated laminates compared to their respective dry laminates.

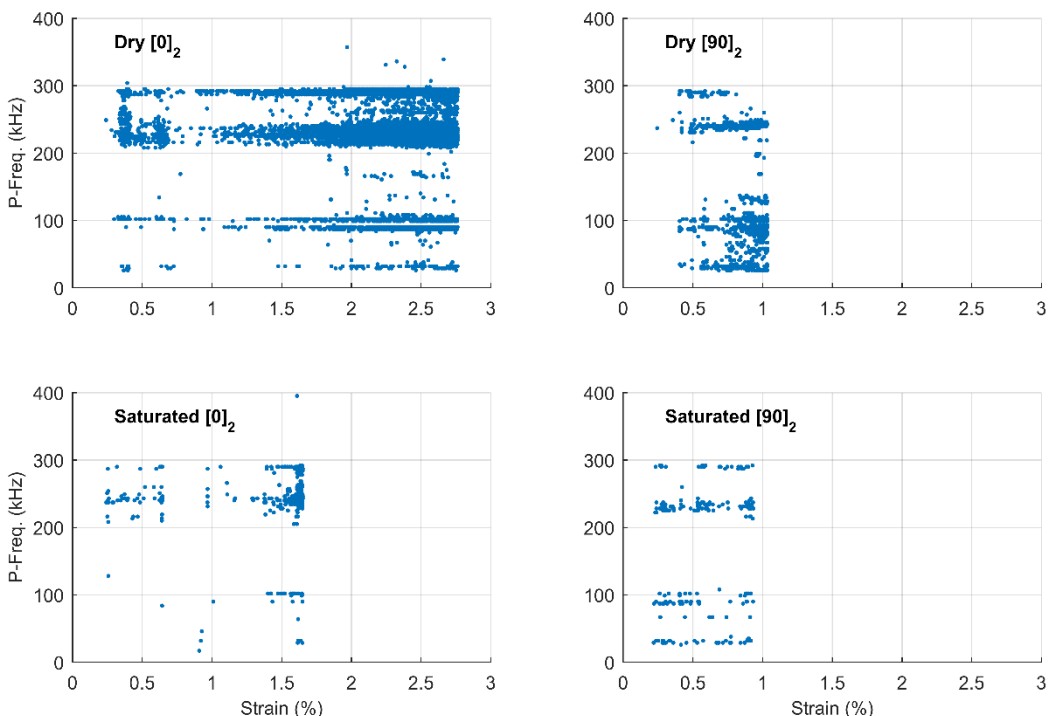

**Figure 10:Typical event Peak-Frequency versus strain for 2-ply laminates**



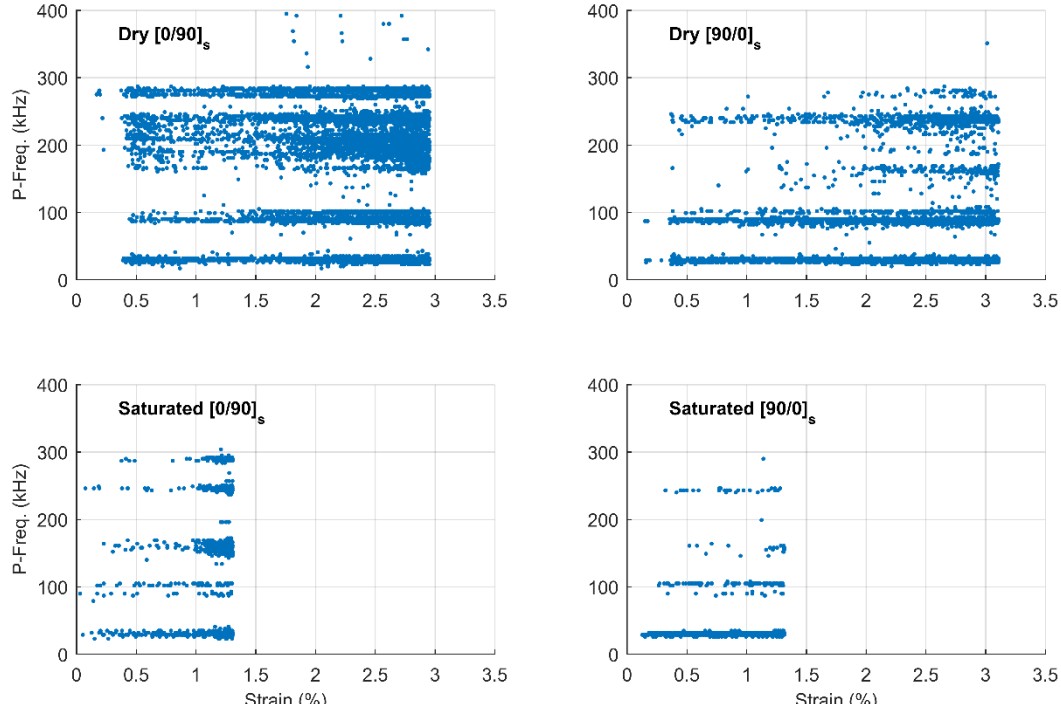

**Figure 11: Typical event Peak-Frequency versus strain for 4-ply laminates**

3.3.2 Event Energy Analysis

The event energies are plotted against strain in Figure 12 and 13 for the unidirectional and cross-ply laminates, respectively.

There is a substantial change in event energies between dry and saturated samples. Acoustic events recorded from dry samples consistently reach $10^8$ aJ while events recorded from saturated samples rarely exceed $10^6$ aJ. As mentioned earlier, this phenomenon has been noticed several times throughout literature. This large reduction in energy makes comparing the event energy response from dry and saturated samples difficult, however the basic trends in AE event energy response of the dry samples can still be discerned in the saturated samples. In the dry $[0]_2$ (Fig 12) there is an initial series of events centered

around 0.5% strain. This peak is also present in the saturated samples, however, the event energies are much smaller. These acoustic events correspond to matrix cracking around the transverse backing tows as referenced above in the peak frequency analysis. In the dry $[0]_2$ tests, acoustic activity picks up again at 1% strain with progressively higher event energies throughout failure. This is also seen in the saturated $[0]_2$ samples but over a much smaller strain range. The progression of event energies for dry and saturated $[90]_2$ laminates also exhibit a similar pattern but with a decrease in event energy as well

as number events in the saturated samples.

The event energy AE results of the cross-ply laminates are shown in Figure 13. Both laminates show an increase in number of events near failure for both conditions. Interestingly, the dry laminates feature many high energy events less than 1%

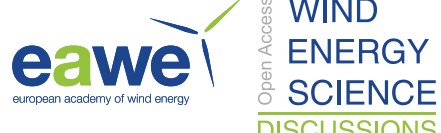

strain that are not present in the saturated samples. The [0/90]$_s$ laminates exhibit a steady trend of increasing event energies throughout the latter half of the damage progression. This trend is not as pronounced in the [90/0]$_s$ laminates.

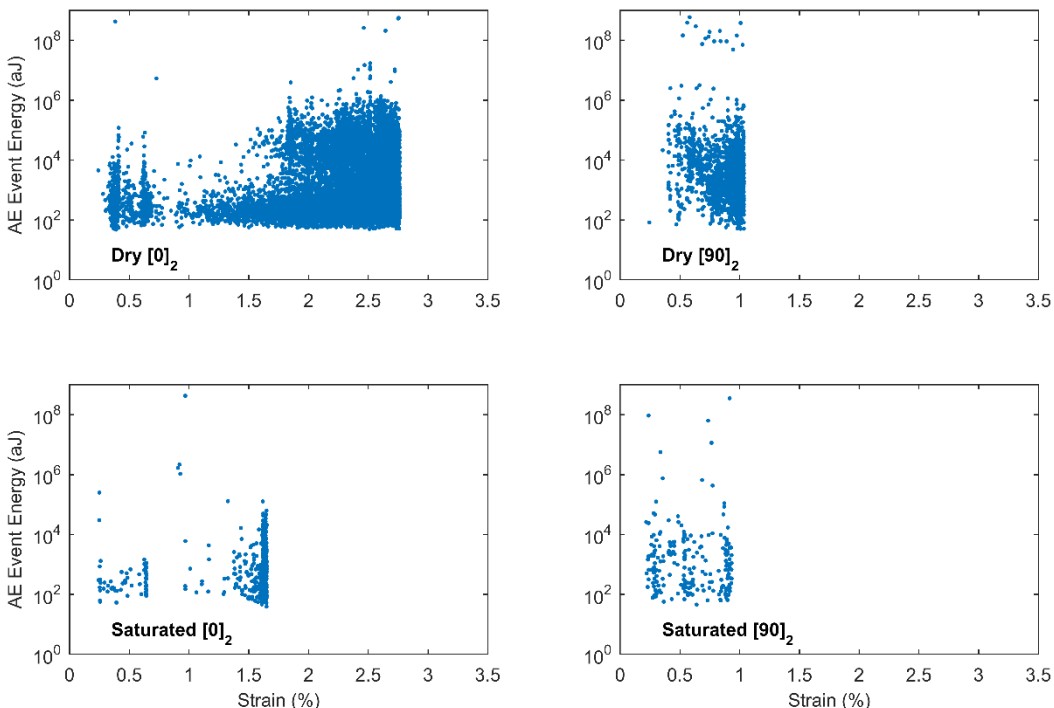

**Figure 12: Typical event energy versus strain for 2-ply laminates:**





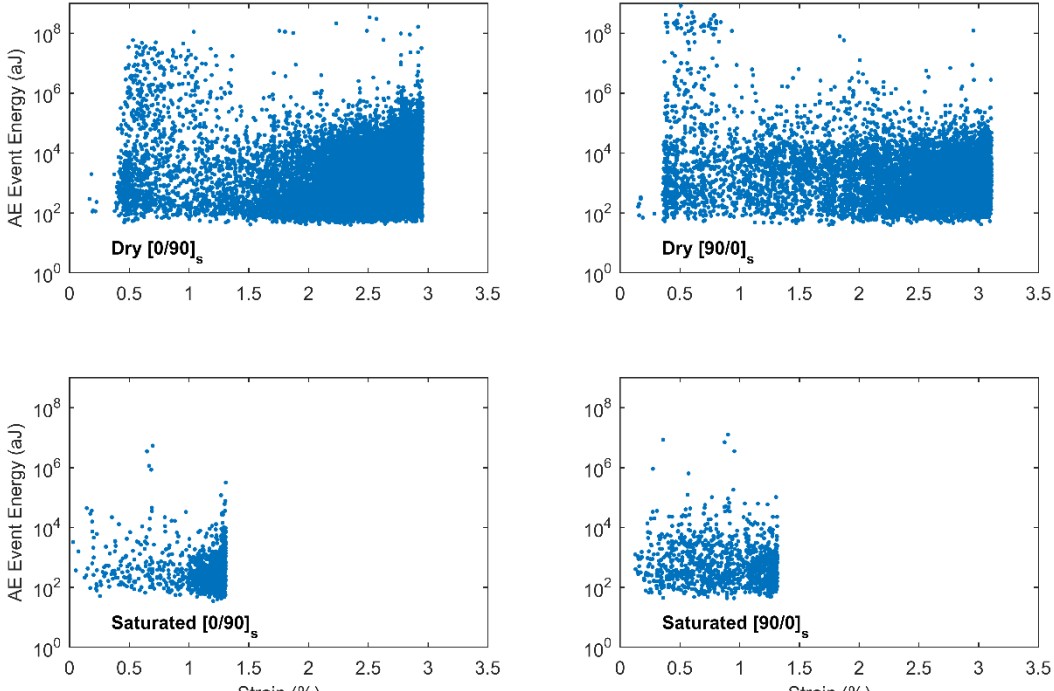

**Figure 13: Typical event energy versus strain for 4-ply laminates**

## 4 Discussion

The substantial reduction in laminate strength (Fig 3) shows moisture uptake to be particularly detrimental for this material
system. Unidirectional laminate strengths were reduced by 40%; cross-ply laminate strengths by 54%. The greater strength
reduction present in [0/90]s and [90/0]s laminates compared to [0]$_2$ laminates indicates that ply interactions significantly
affect damage tolerance of the material system. Ultimate loads emphasize this effect as the saturated cross-ply (containing
equivalent number of 0° plies) laminates failed at a lower unit load than the unidirectional [0]$_2$.

The variation in damage tolerance can partially be explained by examination of damage initiation. Changes in damage
initiation were most evident in the [90]$_2$ layup, shown by both the knee in the stress-strain response (Figure 5) and earlier
event accumulation in the AE response (Fig 10 &12).  Damage in transverse plies is a matrix dominated behavior, as
transverse cracks propagate through resin-rich regions in between fiber tows seen in Figure 6. Lower strain at initiation of
transverse failures in [90]$_2$ samples indicates degradation in the matrix due to moisture absorption. This behavior translated
to cross-ply laminates, where failure in the transverse plies caused deviation in the stress-strain response at similar strain
values of the [90]$_2$ damage initiation (Figure 5). Again, the lowered damage initiation strain was reflected in the AE response
as activity began at lower strains in saturated cross-ply laminates than their dry counterparts (Fig.11 &13). [0]$_2$ samples, in





which transverse cracking is not a dominate mechanism, showed a negligible change in damage initiation strain with aging. It appears that the earlier initiation of damage in the transverse plies due to moisture absorption significantly affects the damage tolerance of the entire composite by introducing cracks and stress concentrations into the primary load bearing plies.

Despite a moisture induced change in damage initiation of the transverse plies, the subsequent damage progressions of dry
and saturated laminates were remarkably similar. Stress-strain and AE response are the tools used to compare damage progression. Stress-strain responses of dry and saturated $[0]_2$ appeared nearly identical up until final failure of saturated $[0]_2$ (Fig. 5). Aside from the slight deviation around 0.3% strain due to premature transverse failures in saturated 90° plies (Fig. 5), the stress-strain response of the dry cross-ply laminates coincided with the saturated laminates indicating little change in the damage progression with moisture.

As mentioned previously, a decrease in AE events and energy has been associated with moisture absorption. This effect was present in this study as well, however, general trends in AE response can still be used for comparison. Dry and saturated samples of both unidirectional and cross-ply laminates experienced the same frequency spectra (Fig. 10 &12) throughout the loading cycle, suggesting that the individual damage mechanisms present were consistent regardless of condition. Event energies (Fig. 11&13), despite being lower in saturated samples, followed similar patterns, pointing to a consistent damage
progression in both dry and saturated samples.

The similar damage progression between dry and saturated sample suggests that the reduced ultimate strength was in fact due to reduced damage tolerance, with moisture uptake increasing sensitivity to damage leading to premature growth. The damage growth attributing to ultimate failure can be observed by inspection of failed coupons. The transverse cracking in $[90]_2$ laminates (Figure 6) showed a lower crack density in saturated samples than in dry, again emphasizing a change in
damage tolerance, with final failure initiating and from a lower damage state in the saturated samples that the dry. This localized damage growth at failure is witnessed in saturated $[0]_2$ and cross-ply (Fig. Figure 7-9) laminates where uniform failures were observed rather than the widespread high energy-release failures seen in dry specimens. In dry cross-ply laminates, damage of load bearing plies was less reactive to stress concentrations induced by failures in transverse plies. Conversely, in saturated cross-ply laminates, damage in transverse plies spawned premature failure in the load bearing plies.
Consistency between the results of both cross-ply layups suggests that this effect is independent of stacking sequence.

## 5 Conclusions

The fiberglass-epoxy system tested in this research experienced significant strength reductions after hygrothermal aging. Unidirectional laminates experienced static strength reductions of 40% while cross-ply laminates experienced a more substantial reduction of 54%. Larger strength reduction in cross-ply laminates compared to unidirectional laminates suggests
the reduction in damage tolerance in a multiangle is not reflected by the lamina behavior, with interacting ply behavior increasing the severity of hygrothermal effects.



Mechanical response of unidirectional and cross-ply laminates, supplemented by acoustic emission data show changes in the damage initiation in aged samples, as well as damage growth at final failure. When saturated, damage in transverse plies initiates at lower strains, however both the stress-strain response and AE response show consistency of subsequent damage progression compared to the unconditioned specimens. Although this data suggests that the damage progression is largely

unaffected by moisture ingress into the composite, it reaffirms that moisture in fact reduces the damage tolerance. Examination of failed test specimens supports a change in damage tolerance with saturated specimens experience more localized damage growth than unaged specimens. The more notably reduction in the strength cross-ply laminates can be explained by an increased sensitivity to damage growth from transverse ply failures. This change in behavior cannot adequately be captured by lamina-based tests, emphasizing the importance of laminate characterization and qualification for

wind energy and MHK application.

**6 Data Availability**

Data is available upon request from the Montana State Composites Research Group.

**7 Competing interests**

The authors declare that they have no conflict of interest.

**8 Acknowledgments**

This research was funded by the Water Power Technologies group at the Sandia National Laboratory.

Assarar, M., Scida, D., El Mahi, A., Poilane, C., and Ayad, R.: Influence of water ageing on mechanical properties and damage events of two reinforced composite materials: Flax-fibres and glass-fibres, Mater Design, 32, 788-795, 10.1016/j.matdes.2010.07.024, 2011.

Bourchak, M., Farrow, I., Bond, I., Rowland, C., and Menan, F.: Acoustic emission energy as a fatigue damage parameter for CFRP composites, Int J Fatigue, 29, 457-470, 2007.

Czigány, T., Mohd Ishak, Z. A., and Karger-Kocsis, J.: On the failure mode in dry and hygrothermally aged short fiber-reinforced injection-molded polyarylamide composites by acoustic emission, Appl Compos Mater, 2, 313-326, 10.1007/BF00568767, 1995.

deGroot, P. J., Wijnen, P. A. M., and Janssen, R. B. F.: Real-time frequency determination of acoustic emission for different fracture

mechanisms in carbon epoxy composites, Compos Sci Technol, 55, 405-412, Doi 10.1016/0266-3538(95)00121-2, 1995.

Garg, A., and Ishai, O.: Characterization of Damage Initiation and Propagation in Graphite/Epoxy Laminates by Acoustic-Emission, Eng Fract Mech, 22, 595-608, Doi 10.1016/0013-7944(85)90123-7, 1985a.

Garg, A., and Ishai, O.: Hygrothermal influence on delamination behavior of graphite/epoxy laminates, Eng Fract Mech, 22, 413-427, http://dx.doi.org/10.1016/0013-7944(85)90142-0, 1985b.

Grammatikos, S. A., Evernden, M., Mitchels, J., Zafari, B., Mottram, J. T., and Papanicolaou, G. C.: On the response to hygrothermal aging of pultruded FRPs used in the civil engineering sector, Mater Design, 96, 283-295, https://doi.org/10.1016/j.matdes.2016.02.026, 2016.

Gutkin, R., Green, C. J., Vangrattanachai, S., Pinho, S. T., Robinson, P., and Curtis, P. T.: On acoustic emission for failure investigation in CFRP: Pattern recognition and peak frequency analyses, Mechanical Systems and Signal Processing, 25, 1393-1407,

https://doi.org/10.1016/j.ymssp.2010.11.014, 2011.





Hamstad, M.: Local characterization of fiber composites by acoustic emission, The Journal of the Acoustical Society of America, 73, 2230-2230, 1983.

Komai, K., Minoshima, K., and Shiroshita, S.: Hygrothermal degradation and fracture process of advanced fibre-reinforced plastics, Materials Science and Engineering: A, 143, 155-166, http://dx.doi.org/10.1016/0921-5093(91)90735-6, 1991.

Kumar, C. S., Arumugam, V., and Santulli, C.: Characterization of indentation damage resistance of hybrid composite laminates using acoustic emission monitoring, Compos Part B-Eng, 111, 165-178, 10.1016/j.compositesb.2016.12.012, 2017.

Liao, K., Schultheisz, C. R., and Hunston, D. L.: Effects of environmental aging on the properties of pultruded GFRP, Compos Part B-Eng, 30, 485-493, Doi 10.1016/S1359-8368(99)00013-X, 1999.

Liu, P. F., Yang, J., and Peng, X. Q.: Delamination analysis of carbon fiber composites under hygrothermal environment using acoustic
emission, J Compos Mater, 51, 1557-1571, 10.1177/0021998316661043, 2017.

Miller, D., Mandell, J. F., Samborsky, D. D., Hernandez-Sanchez, B. A., and Griffith, D. T.: Performance of composite materials subjected to salt water environments, 2012 AIAA SDM Wind Energy Session, 2012.

Mourad, A.-H. I., Abdel-Magid, B. M., El-Maaddawy, T., and Grami, M. E.: Effect of Seawater and Warm Environment on Glass/Epoxy and Glass/Polyurethane Composites, Appl Compos Mater, 17, 557-573, 10.1007/s10443-010-9143-1, 2010.

Ni, Q.-Q., and Iwamoto, M.: Wavelet transform of acoustic emission signals in failure of model composites, Eng Fract Mech, 69, 717-728, https://doi.org/10.1016/S0013-7944(01)00105-9, 2002.

Nogueira, P., Ramirez, C., Torres, A., Abad, M. J., Cano, J., Lopez, J., Lopez-Bueno, I., and Barral, L.: Effect of water sorption on the structure and mechanical properties of an epoxy resin system, J Appl Polym Sci, 80, 71-80, Doi 10.1002/1097-4628(20010404)80:1<71::Aid-App1077>3.0.Co;2-H, 2001.

Nunemaker, J. D.: Effects of saltwater saturation on the static strength and acoustic emission signatures of epoxy glass composites SAMPE Conference Proceedings, Long Beach, CA, 2016.

Nunemaker, J. D.: Static strength reduction and acoustic emission analysis of fiberglass-epoxy samples subjected to various levels of moisture absorption SAMPE Conference Proceedings, Seattle, WA, 2017.

Pashmforoush, F., Fotouhi, M., and Ahmadi, M.: Acoustic emission-based damage classification of glass/polyester composites using
harmony search k-means algorithm, Journal of Reinforced Plastics and Composites, 31, 671-680, 10.1177/0731684412442257, 2012.

Ramirez-Jimenez, C. R., Papadakis, N., Reynolds, N., Gan, T. H., Purnell, P., and Pharaoh, M.: Identification of failure modes in glass/polypropylene composites by means of the primary frequency content of the acoustic emission event, Compos Sci Technol, 64, 1819-1827, 10.1016/j.compscitech.2004.01.008, 2004.

Samborsky, D., Mandell, J., and Miller, D.: The SNL/MSU/DOE Fatigue of Composite Materials Database: Recent Trends, in: 53rd
AIAA/ASME/ASCE/AHS/ASC Structures, Structural Dynamics and Materials Conference, Structures, Structural Dynamics, and Materials and Co-located Conferences, American Institute of Aeronautics and Astronautics, 2012.

Siriruk, A., and Penumadu, D.: Degradation in fatigue behavior of carbon fiber-vinyl ester based composites due to sea environment, Compos Part B-Eng, 61, 94-98, 10.1016/j.compositesb.2014.01.030, 2014.

Sun, P., Zhao, Y., Luo, Y. F., and Sun, L. L.: Effect of temperature and cyclic hygrothermal aging on the interlaminar shear strength of
carbon fiber/bismaleimide (BMI) composite, Mater Design, 32, 4341-4347, 10.1016/j.matdes.2011.04.007, 2011.

Suresh Kumar, C., Arumugam, V., and Santulli, C.: Characterization of indentation damage resistance of hybrid composite laminates using acoustic emission monitoring, Composites Part B: Engineering, 111, 165-178, https://doi.org/10.1016/j.compositesb.2016.12.012, 2017.

Surgeon, M., and Wevers, M.: Modal analysis of acoustic emission signals from CFRP laminates, Ndt&E Int, 32, 311-322, Doi 10.1016/S0963-8695(98)00077-2, 1999.

Suzuki, M., Nakanishi, H., Iwamoto, M., Jiao, G.-Q., Koike, K., Imura, M., Shigemitsu, S., and Jinen, E.: Fatigue Fracture Mechanism of Class A-SMC by Acoustic Emission Method, Journal of the Society of Materials Science, Japan, 36, 1402-1408, 10.2472/jsms.36.1402, 1987.

Tsai, Y., Bosze, E., Barjasteh, E., and Nutt, S.: Influence of hygrothermal environment on thermal and mechanical properties of carbon fiber/fiberglass hybrid composites, Compos Sci Technol, 69, 432-437, 2009.

Tsenoglou, C. J., Pavlidou, S., and Papaspyrides, C. D.: Evaluation of interfacial relaxation due to water absorption in fiber-polymer composites, Compos Sci Technol, 66, 2855-2864, 10.1016/j.compscitech.2006.02.022, 2006.

Voth, M. M.: Exploring frequency based analysis methods for damage identification in fiberglass-epoxy composite systems, SAMPE Conference Proceedings, Seattle, WA, 2017.

Zhou, J. M., and Lucas, J. P.: Hygrothermal effects of epoxy resin. Part I: the nature of water in epoxy, Polymer, 40, 5505-5512, Doi
10.1016/S0032-3861(98)00790-3, 1999a.

Zhou, J. M., and Lucas, J. P.: Hygrothermal effects of epoxy resin. Part II: variations of glass transition temperature, Polymer, 40, 5513-5522, Doi 10.1016/S0032-3861(98)00791-5, 1999b.