# Peer review of "Effects of moisture absorption on damage progression and strength of unidirectional and cross-ply fiberglass-epoxy composites"

_Wind Energy Science, 2018_

## Short Comment (SC1) · 19 Feb 2018

the paper is interesting, however, more effects of moisture about test and experimental sides should be added.

---

## Referee Comment (RC1) · Anonymous Referee #1 · 20 Feb 2018

**1   General comments**

The paper is well written and the topic is quite relevant. Hygrothermal aging of composites still has a significant number of open questions despite many years of existing research. The combination of techniques employed by the authors and the use of different layups allow for a number of interesting clues about hygrothermal degradation to be obtained. However, a number of improvements (listed below) would help augment the scientific value of the paper and make it more suitable for publication.

[Figure]

**2 Specific comments**

- Page 2, line 5: The authors avoid aging at very high temperatures in order to avoid chemically degrading the resin. However, it is also important to acknowledge that aging at 50C can accelerate the chemical degradation of the interface (also mentioned by the authors in line 15). Even avoiding exceedingly high temperatures, it is still risky to assume that specimens saturated at room temperature and 50C have the same degradation level.

- Page 4, line 5: Which tab material did the authors apply? If the tabs also take water this would affect the results of Figure 1.

- Page 4, Fig. 1: The plateau in the diffusion curves mentioned by the authors is difficult to see. In particular, the time interval between the last two measurements seems quite short. How can the authors be sure saturation was reached? Also, did the authors perform an initial drying of the samples before conditioning? If the samples had some initial amount of water when immersion started, the values for $m_\infty$ might be higher than measured. Such an initial drying would also be important to ensure samples labeled as 'dry' provide a moisture-free reference for the study. If not, the authors could label the 'dry' samples as 'as-manufactured' so the reader has a clear idea of the sample condition upon testing.

- Page 8, line 7: Would reorientation of the backing fibers towards the loading direction be another possible explanation for this gain in stiffness?

- Page 13, lines 10-12: Where in the peak frequency analysis did the authors correlate a certain frequency to matrix damage? In general, which frequencies should one expect for each damage type? Is there a clear pattern across material systems and specimen types? The unfamiliar reader is not capable of discerning any information about failure other than the number of events when looking at

Figures 10 and 11 alone. Perhaps a slightly extended discussion on the subject would be interesting.

- Page 15, Discussion: One important degradation driver missing from this discussion that helps explain the observed changes in strength and failure strain is the presence of differential swelling stresses. Since the fibers do not take water and the resin around them swells, significant stress concentrations are created which might help explain the lower failure strain observed in saturated specimens. Furthermore, experimental evidence suggests that resin degradation (at this temperature mainly through plasticization) would not translate to a reduced failure strain, but rather to proportional reductions in stiffness and strength.

**3 Technical corrections**

- Page 4, Equation 1: The subscript in $m_\mathrm{i}$ should be written upright, since it is not a variable.

- Page 5, line 5: Please use mm/min and mm instead of in/min and inch in order to keep the units consistent with Figure 2.

---

## Referee Comment (RC2) · Anonymous Referee #2 · 7 Mar 2018

This is a well written paper with everything to recommend it for publication.

The use of AE monitoring for the mechanical tests adds good information to help characterise the effect of hygrothermal aging on damage mechanisms. As the authors point out in their Introduction (1.3), although considerable previous research exists there is more to be explored and understood here.

Page 2, line 24-25: ". . .AE response associated from (with?) damage behaviour from effects of moisture. . ."

The specimens are testing in static loading. For wind energy blade materials it is dynamic loading effects (fatigue properties) that are most often in focus and it would

have been nice to have some discussion and reference to this.

The AE setup and analysis is well chosen and described here.

The coupon failures for different lay ups and dry/aged condition is very interesting and nicely described. However with the text here broken up by large (and valuable!) images it is not easy to quickly form an overview of the relevant differences. Perhaps a small table can help?

Laminate 1,2,3,4 Dry/Aged Crack density/uniformity "Brooming"failure/"neat" failure and so on...

Again the AE output seems to be suitable for a small table to collate the analysis text...

---

## Author Comment (AC1) · 29 Mar 2018

**1 General Comments**

The authors thank the reviewer for the thorough review of the paper. We feel your comments will create a better publication, and that we can address all of the specific comments with modifications to the final manuscript. A brief comment on each of the specific comments is provided below, along with potential additions to the final manuscript.

**2 Specific Comments**

Point 1- The authors have data on this material system where the coupons were aged at in the 50 °C oven, but not in water. This data shows no effect from the elevated temperature aging. This data has not yet been published, but is pending a conference proceeding. A reference to this data can support this comment.

Point 2 - Saturation curves were measured from witness coupons, without tabs. The final manuscript can clarify this.

Point 3 - As the volume fraction for all coupons are similar, the 2-ply and 4-ply coupons should achieve the same uptake level, simply at different times. This fact supported the decision that saturation was complete. A more descriptive initial condition will be added to the final manuscript.

Point 4 - The author's do not believe the increased stiffness is from reorientation. Rather, the drop in damage tolerance in the saturated coupons does not allow for the transfer of load into the remaining backing fibers, therefore eliminating the stiffness (and added strain) increase.

Point 5 - The authors have an extensive literature review into this very topic, and we will add it into the final manuscript.

Point 6 - The authors agree with the assessment of the reviewer on the effect of the swelling stress on material strength. A discussion of this concept will be added to the final manuscript.
* * *

---

## Author Comment (AC2) · 29 Mar 2018

**1 General Comments**

The authors thank the reviewer for the comments on the manuscript. We believe the comments made will make for an improved final version. A brief response on each of the specific comments from the reviewer is addressed below, and will be added into a final version of the manuscript.

**2 Specific Comments**

Comment 4 - This manuscript is certainly centered on static loading, which is only a

small picture of the wind blade design consideration. A discussion and reference to this implication will be added into the final manuscript.

Comment 6, 7, 8 - The addition of tables to summarize the failure and AE data would be a great addition to the manuscript. They will be added to final manuscript.

---

## Author Response (AR1)

**WES Submission: Effects of moisture absorption on damage progression and strength of unidirectional and cross-ply fiberglass-epoxy composites**

**Reviewer comments and respective revisions summary**

**Anonymous Referee #1**

RC1-1: Page 2, line 5: The authors avoid aging at very high temperatures in order to avoid chemically degrading the resin. However, it is also important to acknowledge that aging at 50C can accelerate the chemical degradation of the interface (also mentioned by the authors in line 15). Even avoiding exceedingly high temperatures, it is still risky to assume that specimens saturated at room temperature and 50C have the same degradation level.

AC1-1: The authors have data on this material system where the coupons were aged at in the 50 ∘C oven, but not in water. This data shows no effect from the elevated temperature aging. This data has not yet been published, but is pending a conference proceeding. A reference to this data can support this comment.

REV1-1: A reference to a work which contains data from isolated temperature aging of the same material system explore in this work. (Page 4, line 9)

RC1-2: Page 4, line 5: Which tab material did the authors apply? If the tabs also take water this would affect the results of Figure 1

AC1-2: Saturation curves were measured from witness coupons, without tabs. The final manuscript can clarify this.

REV1-2: Changes were addressed in page 4 lines 6-7 and 12-13

RC1-3: Page 4, Fig. 1: The plateau in the diffusion curves mentioned by the authors is difficult to see. In particular, the time interval between the last two measurements seems quite short. How can the authors be sure saturation was reached? Also, did the authors perform an initial drying of the samples before conditioning? If the samples had some initial amount of water when immersion started, the values for $m_\infty$ might be higher than measured. Such an initial drying would also be important to ensure samples labeled as 'dry' provide a moisture-free reference for the study. If not, the authors could label the 'dry' samples as 'as-manufactured' so the reader has a clear idea of the sample condition upon testing.

AC1-3: As the volume fraction for all coupons are similar, the 2-ply and 4-ply coupons should achieve the same uptake level, simply at different times. This fact supported the decision that saturation was complete. A more descriptive initial condition will be added to the final manuscript.

REV1-3: The "as-manufactured" descriptor was added to the description of the "dry" sample condition, and the implications of this effect on the moisture content was discussed. A more detailed explanation of measuring the moisture content and verifying the bulk uptake was added (page 4: lines 9-21).

RC1-4: Page 8, line 7: Would reorientation of the backing fibers towards the loading direction be another possible explanation for this gain in stiffness?

AC1-4: The author's do not believe the increased stiffness is from reorientation. Rather, the drop in damage tolerance in the saturated coupons does not allow for the transfer of load into the remaining backing fibers, therefore eliminating the stiffness (and added strain) increase.

REV1-4: the potential for backing fiber realignment was added as a potential explanation for the increase in stiffness. (page 8 lines 9-12)

RC1-5: Where in the peak frequency analysis did the authors correlate a certain frequency to matrix damage? In general, which frequencies should one expect for each damage type? Is there a clear pattern across material systems and specimen types? The unfamiliar reader is not capable of discerning any information about failure other than the number of events when looking at Figures 10 and 11 alone. Perhaps a slightly extended discussion on the subject would be interesting.

AC1-5: The authors have an extensive literature review into this very topic, and we will add it into the final manuscript.

REV1-5: A more detailed explanation of the frequency analysis method and lit review was added in the results section (Peak- frequency analysis section)

RC1-6: Page 15, Discussion: One important degradation driver missing from this discussion that helps explain the observed changes in strength and failure strain is the presence of differential swelling stresses. Since the fibers do not take water and the resin around them swells, significant stress concentrations are created which might help explain the lower failure strain observed in saturated specimens. Furthermore, experimental evidence suggests that resin degradation (at this temperature mainly through plasticization) would not translate to a reduced failure strain, but rather to proportional reductions in stiffness and strength.

AC1-6 : The authors agree with the assessment of the reviewer on the effect of the swelling stress on material strength. A discussion of this concept will be added to the final manuscript.

REV1-5: A discussion of the potential effects of swelling on the strength and damage behavior was added at the end of the discussion section (page 19)

**Anonymous Referee #2**

RC2-4: Page 2, line 24-25: ". . .AE response associated from (with?) damage behavior from effects of moisture. . ." The specimens are testing in static loading. For wind energy blade materials it is dynamic loading effects (fatigue properties) that are most often in focus and it have been nice to have some discussion and reference to this.

AC2-4: This manuscript is certainly centered on static loading, which is only a small picture of the wind blade design consideration. A discussion and reference to this implication will be added into the final manuscript.

REV2-5-7: The importance of static tests in terms of hygrothermal characterization was added to the introduction section (page 2 lines 8-10) as well as the broader impacts of this work in the discussion section

RC2-5-7: The coupon failures for different lay ups and dry/aged condition is very interesting and nicely described. However with the text here broken up by large (and valuable!) images it is not easy to quickly form an overview of the relevant differences. Perhaps a small table can help? Laminate 1,2,3,4 Dry/Aged Crack density/uniformity "Brooming"failure/"neat" failure and so on... Again the AE output seems to be suitable for a small table to collate the analysis text. . .

AC2-5-7: Comment 6, 7, 8 - The addition of tables to summarize the failure and AE data would be a great addition to the manuscript. They will be added to final manuscript.

REV2-5-7: Tables were added to summarize the observations described in the failure analysis sections as well as the acoustic emission section. (Page 10 and 17)